# Genetic Signature and Serocompatibility Evidence for Drug Resistant *Campylobacter jejuni*

**DOI:** 10.3390/antibiotics11101421

**Published:** 2022-10-17

**Authors:** Rozan O. Al-Khresieh, Hala I. Al-Daghistani, Saeid M. Abu-Romman, Lubna F. Abu-Niaaj

**Affiliations:** 1Department of Medical Laboratory Sciences, Faculty of Sciences, Al-Balqa Applied University, Al-Salt 19117, Jordan; 2Department of Medical Laboratory Sciences, Faculty of Medical Allied Sciences, Al-Ahliyya Amman University, Amman 19328, Jordan; 3Department of Biotechnology, Faculty of Agricultural Technology, Al-Balqa Applied University, Al-Salt 19117, Jordan; 4Department of Agricultural and Life Sciences, John W. Garland College of Engineering, Science, Technology and Agriculture, Central State University, Wilberforce, OH 45384, USA

**Keywords:** campylobacteriosis, *Campylobacter jejuni*, multidrug-resistant *Campylobacter*, foodborne pathogens, gastrointestinal illness

## Abstract

Campylobacteriosis, a foodborne illness, is one of the world′s leading causes of gastrointestinal illness. This study investigates the link between human campylobacteriosis and the consumption of potentially contaminated food with *Campylobacter jejuni.* Three hundred sixty samples were collected from humans, chicken cloaca, raw chicken meat, unpasteurized milk, and vegetables. The chickens were obtained from licensed and non-licensed slaughterhouses, and only the necks and wings were studied. Samples were enriched under microaerobic conditions then cultured on the modified charcoal cefoperazone deoxycholate agar. Bacteria was identified by staining, biochemical testing, and molecular identification by the polymerase chain reaction for the virulence genes; *hipO*, *asp*, *dnaJ*, *cadF*, *cdtA*, *cdtB*, and *cdtC*. The genomic homogeneity of *C. jejuni* between human and chicken isolates was assessed by the serological Penner test and the pulse field gel electrophoresis (PFGE). *Campylobacter* was not detected in the vegetables and pasteurized milk, though, only twenty isolates from chickens and clinical samples were presumed to be *Campylobacter* based on their morphology. The biochemical tests confirmed that five isolates were *C. coli*, *and* fifteen isolates were *C. jejuni* including two isolates from humans, and the remaining were from chickens. The colonization of *C. jejuni* in chickens was significantly lower in necks (6.66%) obtained from licensed slaughterhouses compared to those obtained from non-licensed slaughterhouses (33.3%). The antimicrobial susceptibility test showed that all identified *C. jejuni* isolates were resistant to antibiotics, and the majority of isolates (53.5%) showed resistance against six antibiotics, though, all isolates were resistant to ciprofloxacin, tetracycline, and aztreonam. The Penner test showed P:21 as the dominant serotype in isolates from humans, necks, and cloaca. The serohomology of *C. jejuni* from human isolates and chicken necks, wings, and cloaca was 71%, 36%, 78%, respectively. The PFGE analysis of the pattern for DNA fragmentation by the restriction enzyme *Smal* showed a complete genotypic homology of *C. jejuni* human isolates and chicken necks compared to partial homology with cloacal isolates. The study brings attention to the need for effective interventions to ensure best practices for safe poultry production for commercial food chain supply to limit infection with foodborne pathogens, including *Campylobacter.*

## 1. Introduction

Campylobacteriosis is the most common foodborne illness globally and is often caused by *Campylobacter jejuni.* The pathogen is transmitted directly or indirectly to humans from contaminated water and food such as raw or undercooked meat, unpasteurized milk, and vegetables. The clinical symptoms of this gastrointestinal illness are watery to severe inflammatory diarrhea, accompanied often by vomiting, fever, and abdominal pain. A potential complication occurs when campylobacteriosis is associated with Guillain-Barré syndrome, an acquired immune-mediated neuropathy [1], with reactive arthritis, or with irritable bowel syndrome [2]. The reports show that cases of foodborne illnesses caused by *C. jejuni* are more than those produced by *Shigella* and enterotoxigenic *E. coli* combined [3,4]. The worldwide rise of the disease has become a concern for human health. The complications can be life-threatening, making it an issue of public safety and contributing to serious economic implications due to the cost of health systems and loss of workforce productivity. The European Food Safety Authority in 2019 estimated that *Campylobacter* affects more than nine million people annually causing an economic loss of €2.4 billion [5]. The USDA and the Center for Disease Control and Prevention (CDC) in 2019 reported around 1.5 million cases of campylobacteriosis in the United States, with an economic loss of more than two billion dollars [6,7]. The National Antimicrobial Resistance Monitoring System (NARMS) reported an increased multidrug resistance (MDR) in *Campylobacter* species, which poses an additional threat to public safety. The MDR in general is suggested to be due to the widespread use of antimicrobials in animal feeds for growth promotion, or/and the result of the uncontrolled use of antibiotics in humans [8,9,10].

In developing countries, *Campylobacter* is one of the foodborne pathogens that is challenging to food safety and public health. The official data on the incidence of cases is limited which impacts the disease epidemiology. Reports show that the prevalence of *C. jejuni* in developing countries is common in toddlers [11] which is a concern. The World Health Organization (WHO) reports that *Campylobacter* infections in toddlers occasionally lead to death [12]. Studies indicate that an improved management of campylobacteriosis needs additional investigation to better understand the disease epidemiology and the association of its outbreaks to food sources. The source of infection is rarely identified; however, case-controlled studies identified some common sources such as raw poultry, unpasteurized milk, direct contact with animals, fecal runoff of domestic animals, and contaminated drinking water [13].

The variation in the epidemiology of campylobacteriosis between developing and developed countries greatly influences the intervention policies to control the spread of infection. In developing countries, the *Campylobacter* infection is sporadic with seasonal and fewer asymptomatic cases, while in developed countries the infection is endemic with a high incidence of asymptomatic cases [14]. The variation is caused by several factors, including the location of the study, the sensitivity of the diagnostic procedures and the biocontrol programs, food handling, the existence of non-licensed slaughtering houses, and the existence of *Campylobacter* reservoirs in the studied populations [15]. The existence of non-licensed slaughterhouses is not uncommon in developing countries. These sites usually receive no routine inspection and operate with poor sanitation and lack of refrigeration due to no access to clean water nor electricity. Such conditions lead to *Campylobacter* transmission from the infected poultry to humans, causing an increased number of campylobacteriosis cases. The Jordanian Ministry of Health investigates multiple food poisoning outbreaks annually, and several studies reported the association between campylobacteriosis and poultry, beef, eggs, milk, and cheese [16].

The existence of *Campylobacter* spp. as high as 10^6^–10^8^ CFU/g stool in livestock and poultry is indicative of their being essential vectors for the pathogen transmission [17,18]. When broilers get infected with *C. jejuni*, the disease rapidly spreads throughout the flock. Unlike humans, the gastrointestinal colonization of this bacteria in poultry is complex as the highest number is in the mucosal crypts of the caeca and, to a lesser extent, in the small intestine. This complex process is influenced by virulence factors that are crucial for bacterial pathogenesis, such as viability in the intestine, colonization, and the ability to evade the host′s defenses [19,20]. These virulence factors mediate the infectious processes and are coded by several genes for adhesion and colonization (*cadF*), invasion (*ciaB*, *pldA*), a chaperone protein which manages various physiological stresses *(dnaJ*), and the other three genes (*cdtA*, *cdtB*, *cdtC)* of the cytolethal distending toxins (CDTs), which can cause DNA damage in the host cells [21].

The proper clinical diagnosis of *Campylobacter* is critical; though, misdiagnosis is common if it relies only on stool culturing, which is the first step for bacterial isolation. This pathogen cannot tolerate drying and often dies during handling and processing; thus, it needs special requirements to grow outside of the body [4]. Accurate diagnosis has become available through molecular and immunoenzymatic assays, which were developed mainly for epidemiological and source-tracking studies of the infection [22]. The pulse field gel electrophoresis (PFGE), an advanced molecular technique, uses a specific restriction enzyme to create DNA fingerprinting, which allows the determination of genotypic homogeneity among bacterial isolates of the same type [23]. The serotyping is based on a specific monoclonal antibody that was developed and improved by either the parallel use of biotyping, phage typing, or both. Two systems can be used; the Penner system is for the detection of the heat-stable (HS) lipopolysaccharide (LPS) antigen, and the Lior system which detects the heat-labile flagellar and outer membrane protein antigens [24]. This study investigates the link of human campylobacteriosis to the consumed food sources using the DNA fingerprint pattern and the LPS serotyping of *C. jejuni*.

## 2. Materials and Methods

### 2.1. Sample Collection

Three hundred sixty samples were collected from different sources (Amman, Jordan) and analyzed for the presence of *Campylobacter* species. All samples other than the clinical ones were processed according to the FDAs Bacteriological Analytical Manual [25] as detailed below. The Bolton broth was prepared per the manufacturer’s instructions and enriched with Bolton broth selective supplement SR0183, and Laked horse blood SR0048 (Oxoid, Thermo Fisher, Hampshire, UK).

### 2.2. Raw Milk Samples

One hundred unpasteurized milk samples were collected: 50 samples from goats, and 50 samples from cows. Sample processing started with adjusting the pH of milk to 7.6, followed by centrifugation at 12,000× *g* for 4 min. The supernatant was discarded, and the pellet was suspended in 1 mL pre-enriched Bolton broth, then the volume was completed to 10 mL by Bolton broth. Each sample was subjected to pre-enrichment and enrichment steps under microaerobic conditions (5% N_2_, 10% CO_2_, and 85% O_2_) using CampyGen bags (Oxoid, Hampshire, UK). A loopful of the mixture was streaked on modified charcoal cefoperazone deoxycholate agar (mCCDA) plates (Oxoid, Hampshire, UK) and incubated under microaerobic conditions at 42 °C for 48 h.

### 2.3. Chicken Samples

Chicken samples were collected from licensed and non-licensed slaughterhouses in Jordan and transported on ice to the laboratory to be processed the following day. Two sets of chickens’ parts were obtained, and each set had 15 chicken necks and 15 wings sorted individually. The first set was from chickens obtained from the licensed slaughterhouses and the second set was from chickens obtained from the non-licensed slaughterhouses. In total, 25 g skin of either necks or wings was processed by deskinning the desired chicken part, then cutting the skin into pieces and placing it in a sterile stomacher bag containing 100 mL pre-enriched Bolton broth and homogenizing it for 2 min. at 240 rpm using a laboratory stomacher 400 circulator (Steward, Sussex, UK). A 10 mL of the mixture was transferred to a sterile tube and incubated for 48 h at 42 °C under the microaerobic atmosphere as mentioned above. An inoculate of each sample was streaked on mCCDA plates and incubated for 48 h at 42 °C under the specified microaerobic atmosphere. Cloacal samples were individually collected before chicken slaughtering using sterile swabs in Cary-Blair medium (Himedia, Maharashtra, India). The swabs then were transported to the laboratory for immediate culturing on mCCDA plates.

### 2.4. Clinical Human Stool Samples

Seventy stool samples were collected from male and female patients attending governmental hospitals in Amman-Jordan with an age range of 6–59 years. The patients were experiencing clinical symptoms of gastroenteritis and bloody diarrhea. A research consent form to collect patients’ stool samples was approved by the Institutional Review Board at Al-Balqa Applied University (approval code: 26/03/01/749). At least 2 g fresh stool was collected in a leak-proof screw cup container with a sterile applicator stick and then inoculated into a Cary Blair Transport Medium (Himedia, India). The samples immediately were transported to the laboratory under aseptic conditions in portable insulated ice box at 6–8 °C. The stool was cultured within 24 h on mCCDA plates that were incubated under microaerophilic conditions for 24–48 h at 42 °C. Further analysis was performed for bacterial identification [26].

### 2.5. Vegetable Samples

One hundred samples (50 lettuces and 50 spinach) were obtained from supermarkets and local markets. A 5 g sample was placed in a sterile stomacher bag containing 10 mL pre-enriched Bolton Broth, then rinsed with a top bench shaker at 25 rpm for 5 min. Pre-enrichment and enrichment steps took place under microaerophilic conditions at 37 °C for 4 h, then at 42 °C for 20–44 h, respectively. The enriched mixture was streaked on mCCDA plates and incubated under microaerophilic conditions for 24–48 h at 42 °C.

### 2.6. Traditional Identification of Campylobacter

The isolated bacterial colonies were identified by Gram staining [27], motility test, and biochemical tests for oxidase [28], catalase [29], hippurate hydrolysis [30], nitrate reduction [31], and the triple sugar iron (TSI) [25]. Two reference bacterial strains were used as a positive control; *C. jejuni* ATCC 33,291 (Jordanian Food and Drug Administration), and *C. coli* ATCC 43,478 (Microbiologic, UK). The confirmation of the presumptive *Campylobacter* colonies was performed by amplifying selective genes by polymerase chain reaction (PCR).

### 2.7. Molecular Identification of Campylobacter

*Campylobacter* isolates were further cultured on Columbia blood agar plates supplemented with 5% sheep blood for 48 h at 42 °C under microaerophilic conditions. The genomic DNA was extracted using Wizard^®^ Genomic DNA Purification Kit (Promega, Madison, WI, USA) as described by the manufacturer. The PCR was performed to distinguish between *C. jejuni* and *C. coli* by targeting two sets of genes. The first set targeted the *hipO* (hippuricase gene) with a primer sequence Fw (5′-GAA GAG GGT TTG GGT GGT G-3′) and Rv (5′-AGC TAGCTT CGC ATA ATA ACT TG-3′); *asp* (aspartokinase gene) with a primer sequence Fw (5′-GGT ATG ATT TCT ACA AAG CGA G-3′) and Rv (5′-ATA AAA GAC TAT CGT CGC GTG-3′). The second set of genes was *cadF* (genus-specific virulence gene) with a primer sequence, Fw (5′-TTG AAG GTA ATT TAG ATA TG-3′), Rv (5′-CTA ATA CCT AAA GTT GAA AC-3′) and *dnaJ* gene with a primer sequence Fw (5′-ATTGATTTTGCTGCGGGTAG, Rv (5′-ATCCGCAAAAGCTTCAAAAA-3′) [32], which manages the physiological stress of *Camplybocater*. The 25 µL PCR reaction was carried out as follows: 12.5 µL GoTaq^®^ Green Master Mix (Promega, Madison, WI, USA USA), 1.5 µL of each primer (10 µM), 5 μL template DNA then increasing the volume to 25 µL by adding 4.5 µL nuclease-free water. The amplification conditions for *cadF*, *hipO*, *asp*, *dnaJ* were an initial denaturalization at 95 °C for 3 min, followed by 45 cycles at 94 °C for 30 s, the specific annealing temperature (Tm) for each primer for 30 s, followed by 72 °C for 1 min, and a final extension at 72 °C for 5 min. The multiplex PCR was performed to target three cytotoxin genes of *Campylobacter jejuni* (*cdt*), namely, *cdtA* with a primer sequence Fw (5′-CCTTGTGATGCAAGCAATC-3′), Rv (5′-ACA CTC CAT TTG CTT TCT G-3′), and *CdtB* with a primer sequence Fw (5′-GTT AAA ATC CCC TGC TAT CAA CCA-3′), Rv (5′-GTT GGC ACT TGG AAT TTG CAA GGC-3′), and *cdtC* with a primer sequence of Fw (5′-CG ATG AGT TAA AAC AAA AAG ATA-3′), Rv (5′-TTG GCATTATAGAA AAT ACA GTT-3′). The amplification reaction in the thermocycler (Bio-Rad, Hercules, CA, USA) started with an initial denaturalization at 95 °C for 3 min followed by 45 cycles of denaturation at 94 °C for 30s using the specific annealing temperature for each primer for 30 s and the extension at 72 °C for 1 min [33].

The integrity of the amplified DNA was determined by gel electrophoresis using a 1.2% gel in Tris-Borate EDTA (TBE) buffer to which ethidium bromide was added. The desired samples were mixed with 6X DNA loading buffer and were loaded in wells. A 100 bp DNA ladder was loaded in the first well. The gel ran for 50 min at 90–100 V, visualized under a UV transilluminator, and photographed with the gel documentation system (Gel Doc 2000) (Bio-Rad, Hercules, CA, USA). The products’ bands were compared to the used DNA ladder.

### 2.8. Antimicrobial Susceptibility Test

The disc diffusion method was used for evaluating the bacterial resistance to antibiotics. An overnight bacterial culture of the isolates in Mueller-Hinton (MH) broth supplemented with sodium pyruvate, sodium metabisulphite, and ferrous sulphate (Oxoid, UK) was used. The culture density was estimated at 0.5 McFarland (absorbance at 600 nm is 0.063), then a swab was streaked onto MH agar supplemented with 5% lysed horse blood (Oxoid, UK). Plates were allowed to dry before distributing the discs containing the different antibiotics on the agar′s surface [34]. The antibiotics were gentamicin, imipenem, ampicillin, at 10 μg, aztreonam and tetracycline at 30 μg, erythromycin (15 μg), and ciprofloxacin (5 μg) (Himedia, India). Plates were incubated at 42 °C for 48 h, then the diameter of the inhibition zone was measured in millimeters (mm) using a digital caliper. Proposed zone diameters for defining the breakpoints cut-off for *Campylobacter* were determined according to the breakpoints proposed by the European Committee for Antimicrobial Susceptibility Testing [35].

### 2.9. Serotyping using Heat-Stable (HS) Lipopolysaccharide Antiserum

An antigen suspension of cell sensitization was prepared according to the manufacturer’s protocol (Denka Seiken CO., LTD, Tokyo, Japan). In brief, a bacterial inoculum was mixed in 250 mL physiological saline, and 250 μL from both extraction reagent-1 and extraction reagent-2 were added, mixed well, and allowed to react for 10 min. The extraction reagent-3 was added (250 μL) to the mixture, stirred, and centrifuged at 7000 rpm for 5 min. The supernatant was collected for use in the antigen sensitization. A fixed chick red blood cells (cRBCs) (1.5%) suspension was prepared by placing 500 μL of cRBCs with an equivalent amount of buffer solution, mixed well then centrifuged at 3000 rpm for 10 min. The supernatant was removed, and the cell pellet was suspended in 500 μL buffer. A drop of each antiserum was added to a microplate well to 25 μL of the sensitized cell suspension. The microtiter plate was incubated for 30 min in a moisturized box before checking for agglutination. A positive result was documented upon observing a spontaneous agglutination distributed evenly on the surface of the tested well compared to the control well, while the appearance of a central dot in the well indicated a negative result. A positive control from the kit was used.

### 2.10. Pulsed-Field Gel Electrophoresis (PFGE)

The genotyping of *Campylobacter jejuni* using PFGE was performed according to the described method [23]. Briefly, the bacteria were grown at 42 °C for 48 h on Colombia agar with 5% defibrinated sheep blood under microaerophilic conditions (10% CO_2_, 5% H_2_, and 85% N_2_). A cell suspension was then prepared in 85% NaCl and the absorbance at 610 nm was adjusted to 0.57–0.82. The preparation of plugs was carried out by adding 20 μL of ProteinaseK to the 400 μL cell suspension mixed with 400 μL 1% SeaKem Gold (SKG) agarose melted in Tris-EDTA (TE) buffer. The mixture was dispensed into the wells of the plug molds. Lysis of cells in the plugs was achieved in cell lysis buffer containing 50 mM Tris, 50 mM EDTA (pH = 8.0), 10% sarcosine, and 0.1 mg of proteinase K/mL at 54 °C followed by washing with 10–15 mL of TE buffer. Plugs were incubated in a water bath at 54–55 °C with continuous shaking for 10–15 min. The washing buffer was removed, and plugs were allowed to stand at room temperature for 5 min before DNA digestion with 1 μL of *Smal* (40 U/μL) at 37 °C. A part of the prepared plugs was loaded into the wells with a 1% SKG agarose gel. The conditions of electrophoresis were the following: an initial switch time of 6.8 s, a final switch time of 35.4 s, and a gradient of 6 V/cm with 120-degree angles for 20 h in 0.5X TBE on CHEF Mapper XA System (Bio-Rad, Oxoid, Hampshire, UK). The gel was visualized, and the bands were converted into dendrogram using Python version 2.6.5 2011.

### 2.11. Statistical Analysis

Data were analyzed using the SPSS software version 19.0 (Chicago, IL, USA). Chi-square tests were conducted to examine the distribution of *Campylobacter* spp. obtained from different sources. The *p* value < 0.05 was considered the cut-off level for statistical significance.

## 3. Results

### 3.1. Traditional Techniques for Identification of Campylobacter Species

None of the milk or vegetable samples showed bacterial growth on the mCCDA plates. Out of the 130 samples of humans and chickens, only 90 showed bacterial growth on mCCDA plates; 25 were from human stool and 65 were from chicken samples. Out of 90 samples, only 20 isolates were presumed to be *Campylobacter* based on the morphology of colonies that was comparable to that of the reference strains. The pinpoint-size translucent to gray colonies had smooth round edges (Figure 1). The bacterial cells were Gram-negative, curved, or spiral bacilli with seagull-wing shapes. To confirm the identification, colonies were sub-cultured on Columbia blood agar. A series of biochemical tests were performed, including catalase, oxidase, nitrate reductase, glucose utilization, H_2_S production, and hippurate. Fifteen isolates from human stool and chickens combined were identified as *C. jejuni*. The remaining five isolates were identified as *C. coli* as they were hippurate negative, however, they produced H_2_S in the TSI test (Table 1).

### 3.2. Molecular Identification of Campylobacter

The PCR was used to confirm the *Campylobacter* species identified prior to biochemical testing. The following two species-specific genes were used: the *hipO* gene specific for *C. jejuni* and the *asp* gene specific for *C. coli*. The PCR analysis indicated that *C. coli* was present in five chicken samples based on the 500 bp band corresponding to the amplification of the *asp* gene. The PCR for all *C. jejuni* isolates showed a product of 700 bp specific to the *hipO* gene (Appendix A). Only two isolates out of the 70 clinical samples were identified as *C. jejuni* (2.85%); however, the bacterial prevalence in chicken parts varies. The colonization of *C. jejuni* was less in samples obtained from licensed slaughterhouses compared to those obtained from non-licensed slaughterhouses. The colonization of *C. jejuni* in necks and wings was 6.66% and 0%, respectively, in chickens obtained from licensed slaughterhouses compared to 33.3% and 6.66% in their counterparts collected from non-licensed slaughterhouses (Table 2). The colonization of *C. jejuni* was in 20% of the cloacal samples.

The expression of the *cadF* gene, indicated by the appearance of a 400 bp amplicon product (Appendix A), was detected in all *C. jejuni* isolates (100%) despite the source of isolation. The 177 bp product indicated the expression of the *dnaJ* gene (Appendix A) in 46% of the poultry isolates compared to 50% in clinical isolates.

The Multiplex PCR was used for the detection of the cytolethal distending toxin (*cdt*) genes, including *cdtA* (370 bp), *cdtB* (495 bp), and *cdtC* (182 bp) (Appendix A). Variation in the occurrence of the studied virulence genes among isolates was noticed. Poultry isolates of *C. jejuni* displayed 100%, 100%, 46%, 92%, 76.9%, 76.9% for the *hipO*, *cadF*, *dnaJ*, *cdtA*, *cdtB*, *cdtC* genes, respectively, in comparison to 100%, 100%, 50%, 100%, 50%, 50% for human isolates, respectively.

### 3.3. Antibiotic Susceptibility Using the Disc Diffusion Method

All fifteen identified *C. jejuni* isolates were sensitive to imipenem as indicated by inhibition zones ranging from 28 to 33 mm. The isolates showed a variable level of sensitivity to ampicillin (amp), gentamicin (gen), and erythromycin (ery) (Table 3). All isolates were resistant to ciprofloxacin (cip), tetracycline (tet), and aztreonam (atm).

A high percentage of multidrug resistance (MDR) was detected in the isolates of chickens and humans. The MDR was categorized into four different antibiograms based on the number of antibiotics the isolates were resistant to (Table 4). The first antibiogram highlights the resistance of 8 isolates (53.3%) to six antibiotics: ciprofloxacin, ampicillin, gentamicin, aztreonam, erythromycin, and tetracycline. The second and third antibiograms highlights the resistance against five antibiotics: four isolates (26.7%) were resistant to ciprofloxacin, ampicillin, aztreonam, erythromycin, and tetracycline, while the other antibiogram highlights the resistance of two isolates (13.3%) against ciprofloxacin, ampicillin, gentamicin, aztreonam, and tetracycline. The fourth antibiogram represents the resistance of one isolate (6.7%) to four antibiotics: ciprofloxacin, ampicillin, erythromycin, and tetracycline.

### 3.4. Serotyping Using HS-LPS Antiserum

The serotyping analysis indicated that *C. jejuni* isolates reacted with at least 14 antisera encompassed by the Penner test. All clinical and chicken isolates shared the Group P:21 serotype, and no reaction was observed in Group I:10, J: 11, K: 12, L: 15, N: 18, Y: 37, Z: 38, Z2: 41, Z4: 45, Z5: 52, Z6: 55 antisera with the isolates from different sources. A high serocompatibility was detected in the human samples and the cloacal samples (78%), followed by human & chicken necks (71%), and between human & wings (36%) (Figure 2).

### 3.5. Pulsed-Field-Gel-Electrophoresis (PFGE)

The PFGE for the ten *C. jejuni* isolates was clustered into 4 profiles (Figure 3). Three isolates were nontypeable. Genetic homology was verified between poultry neck and human isolates. These are lane 9, lane 7 (human), and lane 6 (neck); partial homology with lane 8, lane 10 (cloaca).

## 4. Discussion

This study investigates the link between sporadic human campylobacteriosis and the consumed food that is potentially contaminated with *Campylobacter jejuni.* Samples were collected from different food sources, and from patients experiencing symptoms of campylobacteriosis. Identification of the isolated *Campylobacter jejuni* was confirmed by molecular and serology assays. This study is the first to report the use of serotyping as a diagnostic tool for *Campylobacter* in Jordan. The culturing and biochemical testing identified two *Campylobacter* spp*.: C. jejuni* and *C. coli.* Out of the 90 chicken samples, 13 isolates were identified as *C. jejuni*, while five isolates were identified as *C. coli.* The findings are comparable to previous studies in terms of the higher prevalence of *C. jejuni* than *C. coli* in chicken samples [36,37], and the relatively low combined contamination level of both bacteria in the samples [38,39]. The study showed a significant higher prevalence of *C. jejuni* in chickens obtained from licensed vs non-licensed slaughterhouses especially in the necks. The C. *jejuni* colonization in the necks was (6.66%) and (0%) in the wings for those chickens obtained from licensed slaughterhouses compared to 33.33% and 6.66% in the necks and wings, respectively, for those chickens obtained from the non-licensed slaughterhouses. The finding is comparable to previous reports for *Campylobacter* species in poultry, especially in chickens raised in non-licensed slaughterhouses: 31.6% in Jordan [40], 34% in Italy [41], 45% in China [38], 29% in Pakistan [42], 17% in Brazil [37], 38% in Germany and 24% in Hungary [43]. The high occurrence of *Campylobacter* in chickens slaughtered at small privately owned facilities or farms, or at non-licensed slaughterhouses is most likely due to improper sanitary procedures and food safety measures. Such practices are not uncommon in developing areas with poor sanitary conditions, limited electricity and refrigeration or where people still have the tradition of consuming undercooked meat. In addition, improper handling and processing of raw or undercooked poultry meat is another major risk factor associated with campylobacteriosis [44]. When infected, poultry becomes a perfect host for *Campylobacter* species, particularly *C. jejuni*, because the body temperature facilitates the growth of *Campylobacter* inside the flock [45]. In addition to poultry, campylobacteriosis is caused by unpasteurized milk, fruits, and vegetables [46,47]. A large-scale cross-sectional study reported raw milk consumption as the main route for *Campylobacter* transmission to humans [48]. This study detected no *Campylobacter* growth from the collected vegetables or unpasteurized milk, which is comparable to prior reports [49,50]. The limited reporting of *Campylobacter* in fruits and vegetables might be caused by a low level of contamination, if any, which makes it difficult to recover the *Campylobacter* from such samples. Overall, poultry meat remains the main attributable source of campylobacteriosis, especially when consumed raw or undercooked. Therefore, overseeing the sanitary conditions where broiler flocks are raised and slaughtered is suggested to be crucial to public health [51] to control the transmission of *Campylobacter* among animals and then to humans.

The current study reported the presence of *C. jejuni*
*in* only 2.8% of the total clinical samples. Previous studies reported *Campylobacter* in clinical samples as low as 1.5% and up to 17%, with a record of 1.5% in Brazil [52], 2% in Sudan [53], 3.8% in India [54], 4.1% in Iran [14], 11% in Pakistan [55], 10% in West Africa [56], and 17% in Algeria [57]. The low percentage of *Campylobacter* in human samples might be due to proper personal hygiene, environmental sanitation, and proper practices for food handling and consumption.

The pathogenicity of *Campylobacter* spp. depends mainly on the existence of virulence genes and the antimicrobial resistance mechanisms they possess. The adhesion of pathogens to mucus membranes, including that of the gastrointestinal tract, is critical for bacterial colonization. It was reported that a mutation in the *cadF* gene leads to the impaired ability of *Campylobacter* to colonize the chicken cecum [58]. Consistent with the results observed by [59], the study reported a high percentage of expression of the *cadF* adhesion gene in isolates of *Campylobacter* from chickens and humans, suggesting its importance in effective colonization. However, the overall results revealed that 50% and 46% of human and chicken samples, respectively, showed no expression of *dnaJ*, a gene supports the survival of *Campylobacter* under diverse physiological stresses [32]. The expression of *cdtA*, *cdtB*, and *cdtC* genes of CDT were assessed using the multiplex PCR assay. CDT is recognized as an important virulence factor as it contributes to DNA damage causing cell death [60]. The three *cdt* genes exist as an operon and their expression controls the activity of the CDT toxin [61]. The data showed that 69% of chickens and 50% of human isolates expressed the *cdt* genes, suggesting a better survival and propagation of the bacteria in chickens. The higher prevalence of the *ctd* genes is comparable to previous findings [62,63]. The low expression of the *cdt* virulence genes among human isolates might be contributed to the small number of clinical samples or the genetic variation in different geographic regions [64].

The evaluation of antibiotic sensitivity of *Campylobacter* was determined. The screening detected the resistance of isolates against several antibiotics such as ciprofloxacin (fluoroquinolone), tetracycline, macrolides and aztreonam (beta-lactam antibiotic). The first two drugs are considered the first choice to treat campylobacteriosis, while aztreonam is considered the most commonly used antibiotic to treat domestic animals in some countries [65]. Previous studies reported a low level MDR in *C. jejuni* isolated from poultry [66,67,68]. The multidrug resistance of pathogens, including *Campylobacter*, is a challenge to health sectors globally [69]. The CDC estimates that over 300,000 infections annually are caused by drug-resistant *Campylobacter* strains, making the treatment a major health problem and causing a financial burden. The development of antibiotic resistance in bacteria might be either due to the misuse of antibiotics in humans, the use of antibiotics in animal feed or in veterinary medicine, or due to the increased industrial waste in the environment [69]. Studies also reported the potential spread of drug resistance from clinical settings to the environment and vice versa [70,71,72]. It is possible that the chickens obtained in this study from non-slaughterhouses were cage-free in areas with contaminated soil, and thus the MDR was developed in the colonized *C. jejuni.*

The current study reports MDR in all of the 15 identified *C. jejuni* isolates from humans and chickens. The antibiotic resistance was categorized into four antibiograms based on the number of antibiotics that the isolated showed resistance to. All isolates were resistant to at least four antibiotics, three of which were ciprofloxacin, tetracycline, and aztreonam. In 53.5% of the isolates, the MDR was up to six antibiotics (cip+ amp+ gen+ atm+ ery+ tet+). The resistance to ciprofloxacin is a concern as this antibiotic is commonly used to treat *Campylobacter* infection. A promising finding, though, was that all isolates were fully susceptible to imipenem, which is used to treat *Campylobacter* infection as it is commonly used in immunocompromised patients and pregnant women [73]; though, it is not approved for animal use [74]. The widespread occurrence of MDR isolates across the poultry farming industry underlines the importance of increased control measures to reduce the bacteria and avoid possible foodborne disease outbreaks.

As *Campylobacter* infection is not among the routine diagnostic schedule in the clinical setting, this might lead to misdiagnosis of many diarrheal cases, and whether the infection is caused by *Campylobacter* strains having MDR from an animal origin. Specific investigation programs to monitor the MDR of *Campylobacter* have been established in developing countries [75]. The current study showed that the *C. jejuni* isolates had a broader range of antibiotic resistance than prior reports in the US, European Union [75], Canada [76], and Lithuania [77]. These reports indicated that the long-term use of tetracycline as a feed additive for poultry was attributed to a large number of tetracycline-resistant strains [78]. The tetracycline gene-*tetO* is the most common resistance gene in *Campylobacter* spp. [79] and can be transferred among bacterial strains [80]. The efflux from TetA and TetB proteins can export tetracycline from the cell, causing tet-resistance [81]. The mutation also leads to MDR in *Campylobacter* as in other bacteria. In *C. jejuni*, mutation can result from the absence of certain genes associated with DNA repairs such as *mutH*, *mutL*, *sbcB*, and *phr* causing bacterial resistance to selective antibiotics [82]. Overall, these findings indicate the need to further control campylobateriosis and to develop a surveillance plan to monitor antibiotic resistance.

The serology assay used to confirm the identity of *C. jejuni* was performed by the Penner system, which is based on the detection of 25 heat-stable (HS) antigens [24]. The human and chicken isolates shared a complete serotype homology for P: 21, while serotypes C: 3 and D: 4, 13, 16, 43, 50 were shared by 93.3% of these isolates. The serotypes F: 6, 17; G: 8; U: 31 were predominant in poultry isolates, with a serohomology of *C. jejuni* between human and chicken samples in 71% for necks, 78% for cloaca, and 36% for wings. The finding is similar to the serotypes reported in clinical samples, bovine and poultry [83,84]. The isolates of *C. jejuni* exhibited a limited HS antigenic variant and thus had no interaction with all the antisera used in the serological examination. The association between human infection and poultry handling most likely is due to the endemic colonization of *C. jejuni* in poultry flocks [85]. The sharing of many HS antigenic determinants suggests a link between poultry food products and human infections.

The genotype assessment for selected virulence genes showed a homology in the PFGE profile between isolates from the human stool samples and isolates from chicken necks compared to their partial homology with the cloaca. Such finding strongly confirms a genetic linkage between chickens and human campylobacteriosis. The colonization is most likely in the gastrointestinal tract of chickens, leading to the contamination of other parts of the animal during processing. The cross-contamination of *C. jejuni* is possible from the chicken cloaca to other parts during slaughtering because of the poor sanitary measures. The finding is comparable to prior reporting on similar genetic relationship between human and poultry meat [23,52,86,87,88].

The comparable PFGE profiles of the *Campylobacter* isolates from humans and chickens propose a possible clonal dissemination suggesting the need to monitor the supplier slaughterhouses to ensure a safe food supply. Further investigation is needed using a larger number of clinical and poultry samples to provide conclusive evidence on the mechanisms of *Campylobacter* colonization in chicken necks and its link to human campylobacteriosis. Overall, this study brings attention to the need for effective interventions to ensure proper practices for chicken production, processing, and safe transportation through the commercial food chain supply.

## 5. Conclusions

The low prevalence of *C. jejuni* in licensed slaughterhouses compared to that exhibited by nonlicensed slaughterhouses highlights the importance of sanitary measures as a critical factor to control bacterial transmission to the public. The high prevalence of MDR phenotype among *C. jejuni* reflects the inappropriate use of antibiotics in poultry farms. The study emphasizes the need to establish specific monitoring programs for animal slaughtering, and for supervised meat transportation to prevent the transmission of *C.*
*jejuni* to humans through the food supply chain.

## Figures and Tables

**Figure 1 antibiotics-11-01421-f001:**
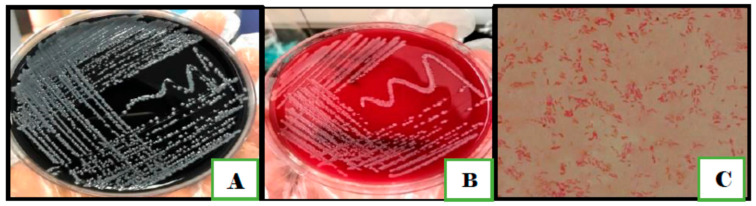
Colony morphology of *Campylobacter* species on (**A**) on mCCDA agar; (**B**) Columbia blood agar; (**C**) Gram staining of *C. jejuni* shows Gram negative spiral rods with seagull-wing shape.

**Figure 2 antibiotics-11-01421-f002:**
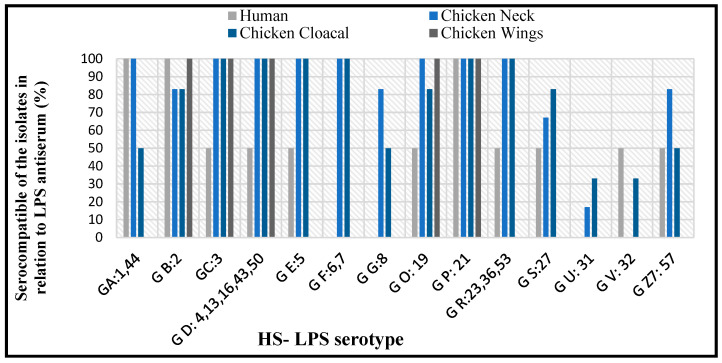
Percentages of serocompatibility in *C. jejuni* among isolates from human and chickens.

**Figure 3 antibiotics-11-01421-f003:**
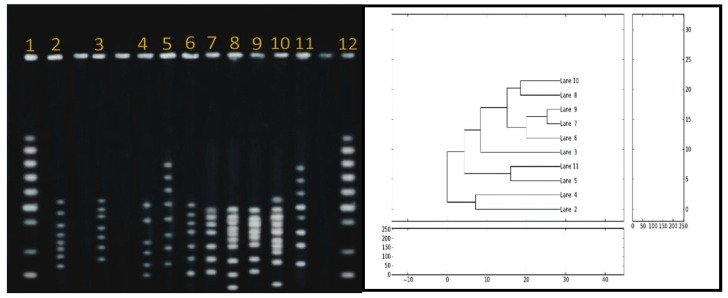
Dendogram with the percentage similarity among the DNA profiles of *Campylobacter jejune* isolated from human and poultry sources. The similarity index was estimated using the Jaccard coefficient and the dendogram was generated through UPGMA method using python version 2.6.5. Lanes 1 and 12, DNA markers; Lanes 7 and 9, human samples; Lanes 2, 3, 5, 6, 11, poultry necks; Lane 4, chicken wings; Lanes 8 and 10, poultry cloaca.

**Table 1 antibiotics-11-01421-t001:** Biochemical tests to differentiate between *Campylobacter jejuni* and *Campylobacter coli*.

Biochemical Tests	*Campylobacter* Species n = 20 Isolates Total	
	*Campylobacter jejuni* n (%)	*Campylobacter coli* n (%)
Catalase test	15 (100%)	5 (100%)
Oxidase test	15 (100%)	5 (100%)
Motility test	15 (100%)	5 (100%)
Nitrate reduction test	15 (100%)	5 (100%)
TSI test		
-Glucose utilization	0 (0%)	0 (0%)
-H2S production	0 (0%)	5 (100%)
Hippurate test	15 (100%)	0 (0%)

**Table 2 antibiotics-11-01421-t002:** The prevalence and the distribution of *Campylobacter jejuni* among human and chicken samples based on the PCR.

Sources		Type of Sample	Samples Colonized with *C. jejuni* n (%)
Animals (n = 90)	Licensed slaughterhouse (n = 30)	Neck (n = 15)	1 (6.66%)
		Wing (n = 15)	0 (0%)
	Non-licensed slaughterhouses (n = 60)	Neck (n = 15)	5 (33.33%)
		Wing (n = 15)	1 (6.66%)
		Cloaca (n = 30)	6 (20%)
Humans (n = 70)	Clinical Samples	Human stool (n = 70)	2 (2.85%)

n = number of samples.

**Table 3 antibiotics-11-01421-t003:** Antimicrobial susceptibility percentages of *Campylobacter jejuni* isolated from poultry and human source to selected antibiotics.

Groups of Antibiotics	Type of Antibiotics	Human *C. jejuni* Isolates n = 2		Chickens’ *C. jejuni* Isolates n = 13	
		Susceptible n (%)	Resistant n (%)	Susceptible n (%)	Resistant n (%)
Fluoroquinolones	Ciprofloxacin (5 μg)	0 (0%)	2 (100%)	0 (0%)	13 (100%)
Beta lactam	Imipenem (10 μg)	2 (100%)	0 (0%)	13 (100%)	0 (0%)
	Azteronam (30 μg)	0 (0%)	2 (100%)	0 (0%)	13 (100%)
	Ampicillin (10 μg)	0 (0%)	2 (100%)	1 (7.6%)	12 (92%)
Aminoglycosides	Gentamicin (10 μg)	0 (0%)	(100%)	5 (38%)	8 (61.5%)
Macrolide	Erythromycin (15 μg)	0 (0%)	2 (100%)	2 (15%)	11 (84.6%)
Tetracycline	Tetracycline (30 μg)	0 (0%)	2 (100%)	0 (0%)	13 (100%)

**Table 4 antibiotics-11-01421-t004:** Antibiograms of *Campylobacter jejuni* isolated from humans and chickens.

Antibiotic Resistance Profiles	Total MDR Isolates (n = 15)		Total Isolates n (%)
	Human Isolates n = 2	Poultry Isolates n = 13	
*cip* + *tet* + *atm* + *amp* + *ery* + *gen* +	1	7	8 (53.3%)
*cip* + *tet* + *atm* + *amp* + *ery* +	1	3	4 (26.7%)
*cip* + *tet* + *atm* + *amp* + *gen* +	0	2	2 (13.3%)
*cip* + *tet* + *atm* + *ery* +	0	1	1 (6.7%)

*cip*, ciprofloxacin; *amp*, ampicillin; *atm*, aztreonam; *gen*, gentamicin; *ery*, erythromycin; *tet*, tetracycline.

## Data Availability

Data is contained within this article or supplementary materials.

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
