# Peer review of "Genetic Signature and Serocompatibility Evidence for Drug Resistant Campylobacter jejuni"

_antibiotics, 2022, doi:10.3390/antibiotics11101421_

Round 1

Reviewer 1 Report

The manuscript entitled "Genetic Signature and Serocompatibility Evidence for Drug Resistant Campylobacter Jejuni" contains lots of data regarding detection of drug resistant C. jejuni, which shows that the authors performed lots of work for this article. However, there many points need to be addressed by the authors to improve the manuscript:

1. The species written on the title needs to be corrected accordingly.

2.The ethical permission is a must for samples collected from human. This article is missing that part.

3. As the authors performed the traditional techniques to identify Camplyobacter from 90 isolates, it would be better if the authors provide the table presenting the results of this experiments as well as the hipO detection in supplementary. It would be great if the authors include the result from the vegetable samples.

4. The writings of gene identity in the text need to be corrected. For example, it should be “hipO” instead of “HipO”, “cadF” instead of “CadF”, etc.

5. The writings of proteins in the text also need to be corrected. For example, it should be “TetA instead of “tetA”, “TetBinstead of “tetB”, etc.

6. The number of isolates are inconsistent, it is written in the result as 90 samples from different sources, while in table 1 there are 90 from animals and 70 from human. Are the isolates from human were not identified using the same method as from animals? Please explain.

7. The tables shown in the manuscript need to reformatted as in the template.

8. In table 3, authors showed the number of isolates that can grow on multiple antibiotics. However, there is no explanation why using this variation and combination of antibiotics.

9. In the conclusion part, the word “present” need to be corrected into “prevent” as I far as I catch to what the authors want to say.

10. C.coli needs a space between C. and coli. And many other typos and grammatical errors present in the manuscript. Please correct them accordingly.

11. The introduction part and the discussion part are, in my opinion, a bit excessive. It would be much better for the reader if the author could make it more compact.

Author Response

Dear Colleague, 

Thank you for dedicating your time to review our manuscript. Your feedback was really appreciated and helpful.  Hopefully all comments will be satisfactory as addressed below:

1. The species written on the title needs to be corrected accordingly. Corrected; thank you.

2.The ethical permission is a must for samples collected from human. This article is missing that part. Added line 156-157; thank you.

  1. As the authors performed the traditional techniques to identify Camplyobacter from 90 isolates, it would be better if the authors provide the table presenting the results of this experiments as well as the hipO detection in supplementary. It would be great if the authors include the result from the vegetable samples.

We added Table 1 to summarize the biochemical tests for all isolates of Campylobacter.

There was no bacterial growth from the vegetables and milk. There was one sentence prior but we emphasized on this results.

About the hipO- in the narrative, we added the percentages of the samples that were amplified.

  1. The writings of gene identity in the text need to be corrected. For example, it should be “hipO”instead of “HipO”, “cadF” instead of “CadF”, etc. Corrected; thank you.
  2. The writings of proteins in the text also need to be corrected. For example, it should be “TetAinstead of “tetA”, “TetB” instead of “tetB”, etc. Corrected; thank you.
  3. The number of isolates are inconsistent, it is written in the result as 90 samples from different sources, while in table 1 there are 90 from animals and 70 from human. Are the isolates from human were not identified using the same method as from animals? Please explain.

We added a clarification in the beginning of the results-the paragraph (Line 275-285) was modified.

  1. The tables shown in the manuscript need to reformatted as in the template. Corrected; thank you.
  2. In table 3, authors showed the number of isolates that can grow on multiple antibiotics. However, there is no explanation why using this variation and combination of antibiotics.

We made it clear that these antibiotics can be used either in the treatment or in feeds

  1. In the conclusion part, the word “present” need to be corrected into “prevent” as I far as I catch to what the authors want to say. Corrected; thank you.
  2. C.coli needs a space between C. and coli. And many other typos and grammatical errors present in the manuscript. Please correct them accordingly. Corrected; thank you.
  3. The introduction part and the discussion part are, in my opinion, a bit excessive. It would be much better for the reader if the author could make it more compact. Some modification to shorten both was done; thank you.

12. We re-reviewed the manuscript for grammar and proofreading. Thank you. 

Reviewer 2 Report

I am very impressed with the presented publication, which is extremely interesting and written in a very good style. 

The study proposed by the authors presents an interesting area of health and medicine news, which causes interest for both scientists and practitioners.

The authors, no doubt, did a good job, including the application of modern methods in this research. The undoubted advantage of the manuscript is the very specific goal and really interesting and reliable introduction of the presented research. 

Author Response

Thank you so much and praise. We appreciate it.

A revised manuscript is attached. 

Reviewer 3 Report

This is a good-written manuscript on a topic that is relevant. I have no serious or substantive comments. I have very few suggestions on how to improve the manuscript:

1.     Line 278 “None of the 278 water, milk and vegetable samples showed bacterial growth on the mCCDA (Figure1).” – it`s better put “(Figure1)” in the end of next sentence.

2.     Check reference N 6

Author Response

We thank you so much and appreciate your feedback. The attached manuscript is for your kind consideration. In addition, here are the responses to your valuable comments: 

  1. Line 278 “None of the 278 water, milk and vegetable samples showed bacterial growth on the mCCDA (Figure1).” – it`s better put “(Figure1)” in the end of next sentence.

We considered the changes, thank you.

  1. Check reference N 6 – thank you so much; the year was corrected.

Reviewer 4 Report

Thanks for your good effort and presentation of work.

  • The author can add the following reference that related to this work “Wafaa A. Abd El-Ghany (2019): One health approach of campylobacteriosis in Egypt: An emerging zoonotic disease. The Journal of Infection in Developing Countries, 13(11): 956-960”.

Author Response

We thank you so much and appreciate your feedback. The attached manuscript is for your kind consideration. In addition, the reference below was added per your request.  

“Wafaa A. Abd El-Ghany (2019): One health approach of campylobacteriosis in Egypt: An emerging zoonotic disease. The Journal of Infection in Developing Countries, 13(11): 956-960”.

Added; thank you so much for your review.

Round 2

Reviewer 1 Report

Dear Authors,

Thank you for your effort to fulfil the suggestions from the reviewers. Now the manuscript is in a much better condition. However, the ethical clearance is still incomplete, instead of only stating that the research consent was approved by the appropriate institution, the authors need to write also the number or serial of the ethical permission issued by the certain institution.

Author Response

Dear Colleague, 

Our apology for any inconvenience this review might have caused to you, but we really appreciate your time and feedback to allow us publish our work. 

The number for the approved IRB for the work was included in the paper (Lines 155-157). We will be uploading the form to the editor for record so it can be shared with you. The revised manuscript is attached for your final approval. Thank you so much. 

Round 3

Reviewer 1 Report

Thank you for your effort in revising the manuscript. It is now publishable.